# Spatial Assessment of Wildfires Susceptibility in Santa Cruz (Bolivia) Using Random Forest

Marcela Bustillo Sánchez [1,†], Marj Tonini [1,*,†], Anna Mapelli [2] and Paolo Fiorucci [2]

1   Faculty of Geosciences and Environment, Institute of Earth Surface Dynamics, University of Lausanne, CH-1015 Lausanne, Switzerland; marcela.bustillo.sanchez@gmail.com
2   CIMA Research Foundation, 17100 Savona, Italy; anna.mapelli@cimafoundation.org (A.M.); paolo.fiorucci@cimafoundation.org (P.F.)
*   Correspondence: marj.tonini@unil.ch
†   These authors contributed equally to this work.

**Abstract:** Wildfires are expected to increase in the near future, mainly because of climate changes and land use management. One of the most vulnerable areas in the world is the forest in central-South America, including Bolivia. Despite that this country is highly prone to wildfires, literature is rather limited here. To fill this gap, we implemented a dataset including the burned area that occurred in the department of Santa Cruz in the period of 2010–2019, and the digital spatial data describing the predisposing factors (i.e., topography, land cover, ecoregions). The main goal was to develop a model, based on Random Forest, in which probabilistic outputs allowed to elaborate wildfires susceptibility maps. The overall accuracy was finally estimated by using 5-fold cross-validation. In addition, the last three years of observations acted as the testing dataset, allowing to evaluate the predictive performance of the model. The quantitative assessment of the variables revealed that "flooded savanna" and "shrub or herbaceous cover, flooded, fresh/saline/brakish water" are respectively the ecoregions and land cover classes with the highest probability of predicting wildfires. This study contributes to the development and validation of an innovative mapping tool for fire risk assessment, implementable at a regional scale in different areas of the globe.

**Keywords:** wildfires mapping; Bolivia; machine learning; model validation; land use; ecoregions; slash-and-burn

## 1. Introduction

Wildfires are perturbing events affecting ecological processes and are part of the dynamics of many ecosystems in the world, influencing their composition, structure, and functioning. These hazards can cause significant losses in terms of vegetation, houses, and human and animal lives [1–3]. Wildfires are becoming more frequent and extended in the last years, mainly under the influence of climate changes [4–6] and land use management [7–10]. Hence, the need to study them through their modelization, in terms of both fire spread and fire risk/susceptibility assessment, and to understand how to limit the disastrous effects they can have on the environment and on the socioeconomic tissue.

A wide range of methods have been developed to model fire hazards, typically taking advantage of Geographical Information System (GIS) and Remote Sensing (RS) techniques. These can be classified into physics-based methods, multi-criteria analyses coupled with statistical approaches, and machine learning methods. Physics-based methods involve the simulation and the prediction of fire behaviors through mathematical equations of fluid mechanics, combustion of canopy biomass, and heat transfer mechanisms [11–15]. Multi-criteria analyses, coupled with statistical methods, assumes that the probability of occurrence of burned areas can be quantitatively assessed by investigating the relationship between fire occurrences and predisposing factors [16–24]. Finally, machine learning (ML) methods are based on more or less complex data-driven algorithms able to model the

hidden and non-linear relationships between a set of topographical and land use/land cover related predisposing factors with the observed wildfire events [25–40].

ML approaches successfully applied for wildfire risk assessment and susceptibility mapping mainly include Artificial Neural Networks (ANN) [27,28,35,37,39], Support Vector Machines (SVM) [26,32,35,37,39], Random Forests (RF) [25,26,29,32,34–37,39,40], and Multiple or Logistic Regression (LR) [25–30,39]. Most of the studies cited above involve the use of different approaches to evaluate which one performs better. For example, in [28] authors compare LR and ANN to model wildfire risk and to detect potential area for fire occurrence; it resulted that ANN had the higher accuracy, estimated as prediction capability. In [27] authors find that Kernel LR model outperforms the benchmark, based on SVM, for tropical forest fire susceptibility mapping in a protected area in Vietnam. In [37] authors evaluate the performances of three ML approaches (RF, SVM, and ANN) for the elaboration of a wildfire susceptibility map in Iran; in this case RF provided the most accurate predictions. Similarly, authors in [32] proved that RF had the highest predictive accuracy compared to SVM in a study in Dayu County (China). In [34] authors compare two stochastic approaches (RF and extreme learning machine) versus a deterministic procedure for wildfire susceptibility assessment in a region in Portugal, revealing the advantage of using ML-based methods, especially RF since it additionally provides the internal evaluation of the variable importance ranking. Several ML methods are compared in [39] to evaluate their potential for wildfire susceptibility mapping in Amol County (Iran); even in this case, RF performed the highest accuracy, followed by SVM, while LR had the lowest accuracy.

Susceptibility is a fundamental concept in wildfire risk management. Broadly speaking, it can be defined as the extent to which a causal mechanism might affect and destabilize a potentially hazardous system [41]. In the spatio-temporal domain, susceptibility maps indicate areas with the potential to experience a particular hazard in the future based solely on the intrinsic local properties of a site and on the observed past events, expressed in terms of relative spatial likelihood. According to Tonini et al. [40], wildfire susceptibility maps display the wildfire's occurrence probability, ranked from low to high, under a given environmental context.

From the above-mentioned literature review, RF has proven to be one of the most effective algorithm for wildfire susceptibility assessment. This is attested by the higher performances in terms of accuracy achieved by RF in comparison with other methods. RF stands out among other ML algorithms because of the following factors: its calibration is quite easy as it involves only a few parameters and the data do not need to be rescaled or transformed; it reduces over-fitting in decision trees and helps to improve the accuracy; it automatically outputs the variable importance ranking; it can handle directly categorical variables, such as land use classes or vegetation type, which are key factors in wildfire susceptibility assessment.

The central-South American forest is one of the areas most affected by wildfires in the world [42,43]. Wildfires risk in the Amazonian forest will probably be higher in the near future because of the increasing frequency of drought periods coupled with the growing rate of deforestation [44–46]. The main cause of fire ignition in the area is human-made, principally due to the commune practice of slash-and-burn [47,48]. This, called in jargon *chaqueo*, consists of cutting trees, low vegetation/agricultural residuals, and burning the biomass to make way for agriculture, livestock, logging, or simply to clear the agricultural land to prepare fields for the next year's crop. This practice can easily get out of control and initiate large fires. For example, in 2019, Bolivia faced an extremely extensive wildfire event that had a serious ecological impact in the department of Santa Cruz [49,50]. This complex area is characterized by a mosaic where wet and dry tropical forests alternate with savannas, and it is extremely prone to wildfires. Despite Bolivia being amongst the top-ten countries with the highest expected annual burned forest area at risk in the world [42], the literature on wildfire's risk and suppression is quite limited, principally because of the scarcity of available data and resources [51]. To fill this gap, as part of the present study, we implemented an accurate dataset of burned areas based on Moderate Resolution Imaging

Spectroradiometer (MODIS) wildfire product and reporting events that occurred in the entire department of Santa Cruz in the period of 2010–2019. The factors that can predispose the wildfires, such as the topography of the area, the land cover, and the ecoregions, were also collected and processed in the form of digital spatial data. This information allowed estimating the susceptibility of wildfires in the entire department, with a special focus on the municipality of San Ignacio de Velasco. Analyses were performed by using an ML-based approach, namely RF, and outputs presented in the form of maps were finally validated. In addition, the influence of the different predisposing factors and the relative probability of prediction success over a range of discrete values, corresponding to the different classes of land cover and ecoregions, were investigated and discussed.

## 2. Study Area

The study area (Figure 1) corresponds to the department of Santa Cruz in Bolivia. It includes 15 provinces and 56 municipalities, with capital Santa Cruz de la Sierra, in the Andrés Ibáñez province. With an area of 370,621 km², Santa Cruz is the is the largest of the Bolivian departments, covering 34% of the entire national territory. It includes part of the Amazonian and the Chaco plains, with an elevation around 800 m.a.s.l. for the large majority of the territory, which can exceed 1500 m.a.s.l. on the sub-Andean relief.

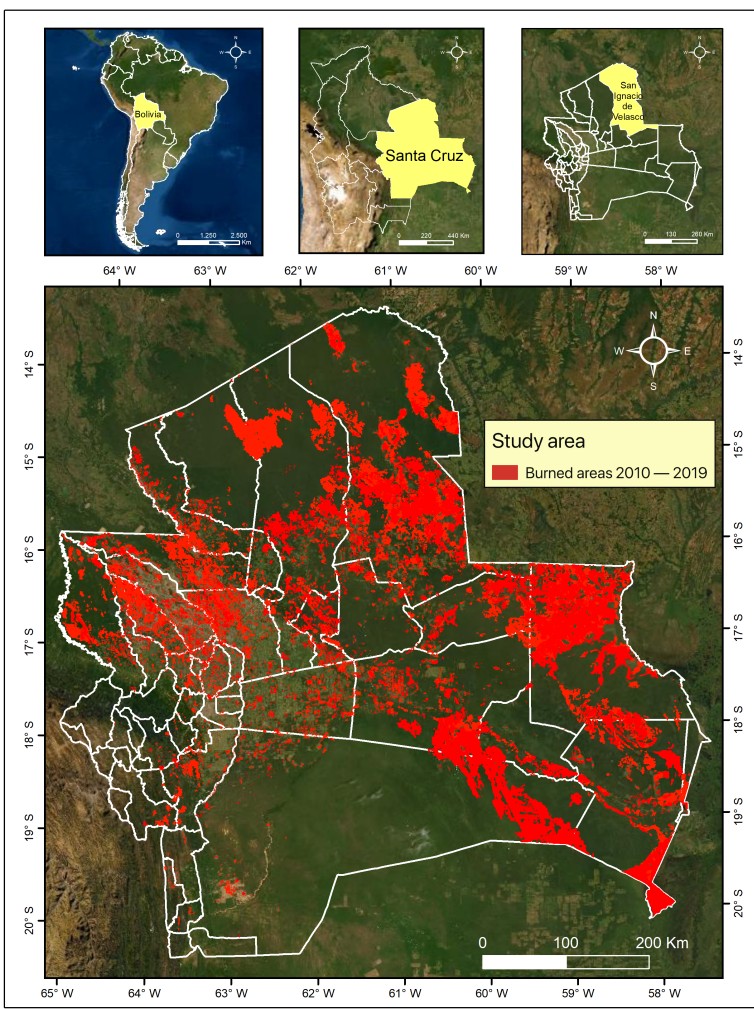

**Figure 1.** Study area: department of Santa Cruz in Bolivia with burned areas (2010–2019, from MODIS). The municipality of San Ignacio de Velasco is also indicated in the top-right map.

Wildfires are historically recognized as part of the disturbance regime of the area. However, in the recent years, due to their increasing frequency and amplitude, and to their

relationship with human activities and agricultural practices, wildfires management and containment has become a challenging task [52]. The main cause of wildfires in Bolivia is the slash-and-burn agricultural practice, followed by activities related to the pasture management, waste burning, hunting, and others human activities [48,51]. According to MODIS data (Figure 1), in the period from 2010 to 2019, about 2203 wildfires per year (Figure 2) resulted in an average burned area of 1,266,275 ha (Figure 3).

In general, the largest number of events occurs between July and October, with August and September being the two months with the highest concentration of burned area. Although forest in this area is exposed to a marked seasonality, and hence it is susceptible to changes in fire regimes, we did not consider this variability in the present study. Indeed, since typically there is only one fire season that hits during the frame-period coinciding with the driest months of the year, monthly burned areas have been aggregated on a yearly basis.

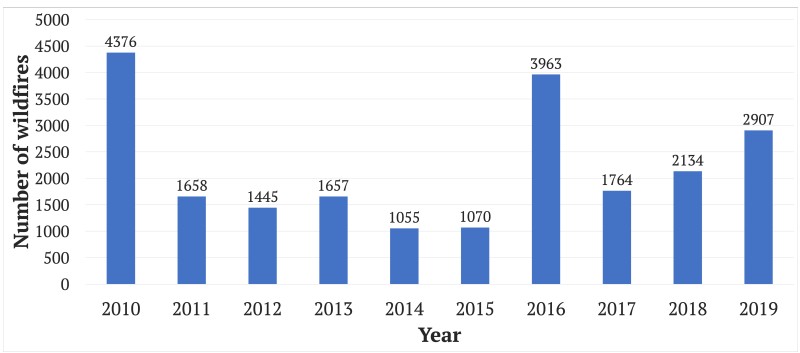

**Figure 2.** Number of wildfires in the department of Santa Cruz (Bolivia) from 2010 to 2019.

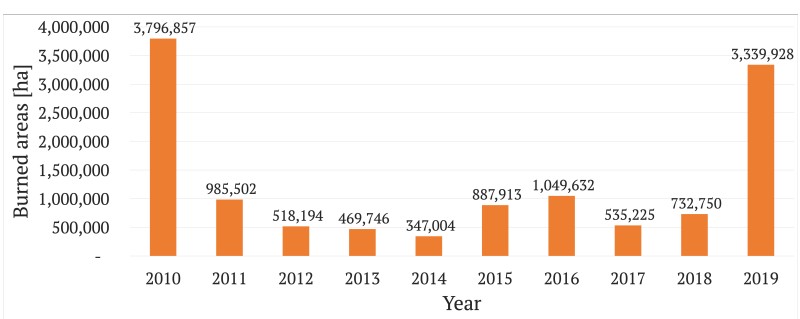

**Figure 3.** Burned areas in the department of Santa Cruz (Bolivia) from 2010 to 2019.

*Municipality of San Ignacio De Velasco*

San Ignacio de Velasco is the largest municipality of the province of José Miguel de Velasco, located at the northeast of the department of Santa Cruz (Figure 1). This area was selected to carry out analyses at a more local scale because it has been the municipality most affected by wildfires within the entire study area during the recent period (2010–2019) (Figure 4) [53]. With a surface of about 48,960 km$^2$, San Ignacio de Velasco has the highest demographic growth after the city of Santa Cruz de la Sierra. Administratively, its area is divided into twelve districts, of which two are highly urbanized, nine are inhabited by indigenous communities, and the last corresponds to Noel Kempff Mercado National Park. The forestry potential and the tourism are the main resources of this municipality.

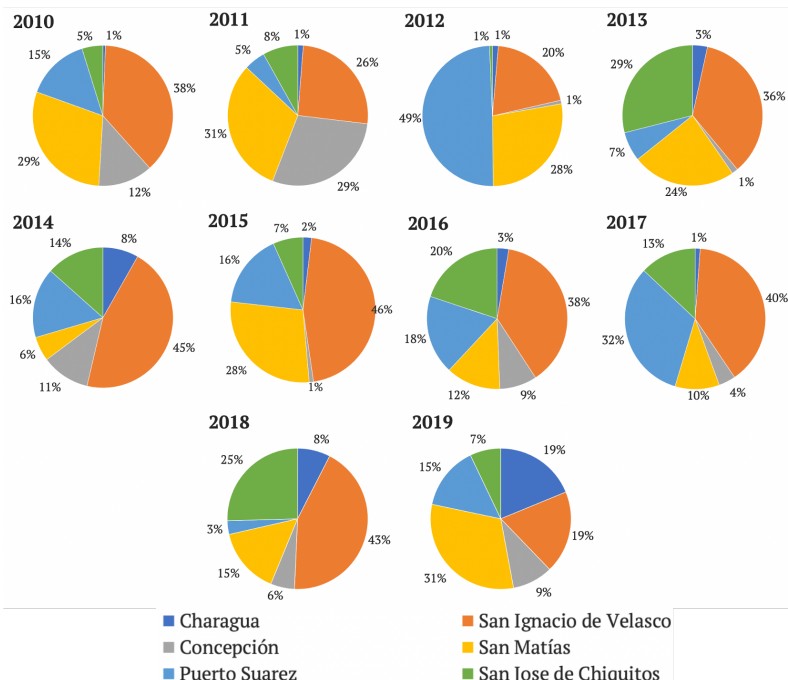

**Figure 4.** Proportion of burned area amongst the six most affected municipalities of the department of Santa Cruz in the period of 2010–2019.

## 3. Materials and Methods

The methodological workflow developed in the present study (Figure 5) includes the following process:

- Implementation of the database, including: (i) the dependent variable (i.e., the burned areas derived by MODIS product); (ii) the independent variables (based on the topographical, ecological, and land cover/vegetation).
- Implementation of an ML approach, using RF and five equal-size folds for the validation procedure, allowing to maximize the spatial generalization of the predictions.
- Elaboration of the wildfire susceptibility maps, based of the probabilistic outputs resulting from RF, and assessment of the variable importance ranking.
- Validation of the performances of the model performed by estimating the Area under the Receiver Operating Characteristic (ROC) curve (AUC), and computed considering the temporal splitting of the original dataset into training (2010–2016) and testing (2017–2019).

### 3.1. Dependent Variable: Burned Areas

Data acquisition and implementation of the input datasets are the most challenging steps in any modeling study. Wildfire prevention, defense, and suppression plans require, first and foremost, accurate estimates of the differential susceptibility of the land to wildfires in relation to the characteristics of the territory and to past observed events. Thus, a key factor for wildfire susceptibility modeling is the observed burned areas, available as mapped fire perimeters and spanning several years.

Burned areas were selected and processed based on the collection of the 6 MODIS burned area mapping product (MCD64), which contains fewer unclassified grid cells as a result of its improvements. In more details, MCD64 employs daily 500-m MODIS surface reflectance data coupled with 1-km MODIS active fire observations [54]. The MODIS burned area product has been downloaded from the University of Maryland (ftp://ba1.geog.umd.edu/, accessed on 20 March 2020) as a GeoTIFF file. Detected burned areas are labeled with the Julian day of the given month for each monthly GeoTIFF. These features have then been aggregated on a yearly basis and the yearly raster datasets clipped over the study area (Figure 1).

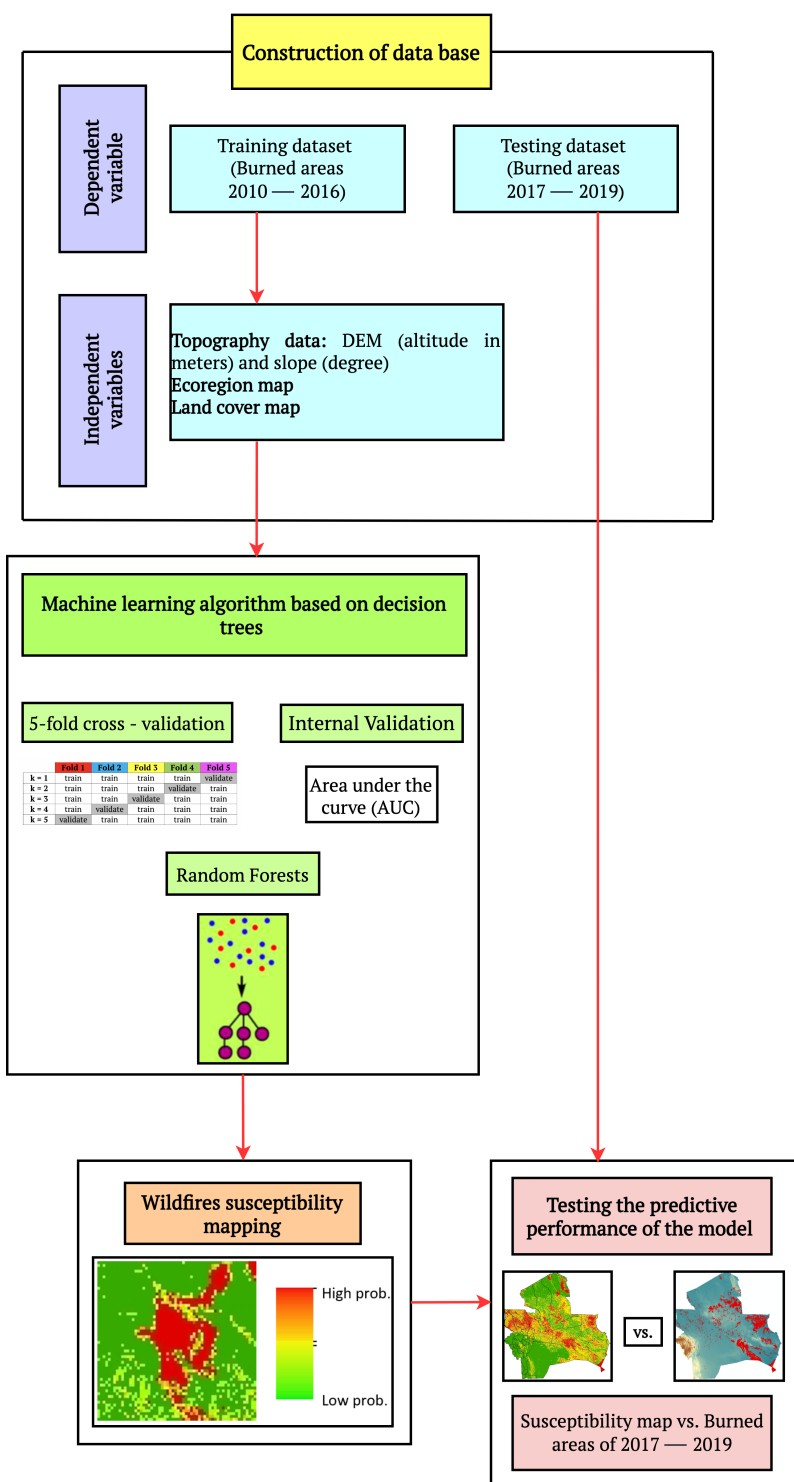

**Figure 5.** Workflow of the methodology employed in this study.

### 3.2. Independent Variables: Predisposing Factors

The following independent variables (Table 1) provide a detailed knowledge of the topography, ecological conditions, and land cover, including vegetation, allowing to understand how these factors can predispose to wildfire occurrence: Digital Elevation Model (DEM) and slope (derivative of DEM), ecoregions (assemblage of species, natural communities, and environmental conditions [55]), and land cover (physical material at the surface of the earth, that includes vegetation information). All of these maps were elaborated in raster format with a grid cell size of $100 \times 100$ m (Figure 6), which is good enough to

capture the spatial characteristics of wildfire locations and, at the same time, large enough to assure a reasonable processing speed.

It is worth noting that meteorological factors, like wind speed and wind direction, temperature, humidity, and rainfall were not included as predisposing factors for this study since, according to Tonini et al. [40], these are local conditions that cause a hazard to occur if and only if the area is susceptible to that hazard (i.e., acting as triggering factors), while the susceptibility is assessed based only on factors that are stable over time (i.e., predisposing factors).

**Table 1.** Summary of predisposing factors and their characteristics.

| Pred. Factors | Variable Type | Range |
|---|---|---|
| DEM | Numerical (meters) | 75–3330 |
| Slope | Numerical (degrees) | 0–76.27 |
| Ecoregions | Categorical | 9 classes |
| Land cover | Categorical | 10 classes |

### 3.2.1. Topographic Conditions: Altitude and Slope

Topography is an important factor that influences wildfires because its properties affect the distribution, composition and flammability of vegetation, local climate (such as average wind speeds), and human accessibility. Therefore the raster DEM, obtained from the competent authorities of the Municipal Autonomous Government of Santa Cruz, played an important role in this study. Slope was extracted based on the DEM; this factor is important since an increase in slope can increase the fire spread rate. Fire can spread more quickly up the steep areas and less quickly down the steep areas.

### 3.2.2. Ecological Conditions: Ecoregions

Ecoregions can be defined as relatively large areas of land or water, containing a distinct assemblage of natural communities and species which share similar environmental conditions. The biodiversity of flora, fauna, and ecosystems differs from one ecoregion to another [56]. This factor is important for the assessment of wildfires in Santa Cruz since it provides information about the vegetation of the area. The information regarding vegetation characteristics of each ecoregion helped in determining which classes have to be prioritized and maintained. The department of Santa Cruz includes nine ecoregions (Table 2), described below [57–59].

- Yungas (also called humid forest): a cloud forest located between 1000 and 3300 m.a.s.l., where permanent moisture is supplied by cloud drizzle and rainfall brought from the Amazon basin by the easterlies. The beta diversity of this ecosystem is the highest in Bolivia.
- Bolivian Tucuman forest: located between 300 and 3300 m.a.s.l. In this ecoregion, the minimum annual temperature range is lower than in Yungas because of the influence of cold southerly winds, called "surazos". The vegetation cover is dense, including trees more than 15 m tall.
- Southwestern Amazon forest: located between 150 and 500 m.a.s.l., it is composed of all the Amazon forest types. The species richness is the same as that of the moist Yungas forest. Trees are more than 45 m tall. This region has suffered from strong human pressure.
- Flooded savanna: located between 100 and 200 m.a.s.l., it is in fact a seasonally flooded savanna due to the numerous rivers from the Andes that flow through the Amazon lowlands.
- Gran Chaco (also called dry forest): located between 200 and 600 m.a.s.l. It has the lowest mean annual precipitation (795 mm), a mean annual temperature of 21.7 °C, and a maximum of 48 °C. It is among the largest and best preserved dry forests in the world.

- Chiquitano Dry forest: located in a transition zone between the moist Amazon rain forest and the Gran Chaco dry forest, at an altitude between 100 and 1400 m.a.s.l. It is endemic to Bolivia, highly biodiverse, and it has been extremely affected by wildfires in recent years.
- Dry Inter-Andean forest: located between 500 and 3300 m.a.s.l., and includes patches of dry forest alternated with Yungas forest and deep inaccessible valleys. Due to its topographical specificity, this ecosystem is characterized by a variety of endemic species.
- Chaco Serrano: is dominated by the horco-quebracho (*Schinopsis hanckeana*) along with the drinking molle (*Lithrea molleoides*), especially in the south, and by a large number of cacti and spiny legumes in the north. At higher altitudes, the forest is replaced by grasslands or gramineous steppes with a predominance of species of the genus *Stipa* and *Festuca*.
- Cerrado: a wide range of climatic conditions exists across the Cerrado ecoregion. Precipitations are between 1000 and 2000 mm per year, with a pronounced dry season from April to September, and mean annual temperatures ranging from 16 °C to 25 °C. This ecoregion is characterized by an enormous biodiversity of plants and animals that is progressively threatened by the expansion of agriculture and the burning of vegetation to make charcoal.

### 3.2.3. Landscape Features: Land Cover

Land cover represents the landscape features on the Earth's surface. The different characteristics (such as load and moisture content) of the distinct land cover types can affect the ignition and spread of fires. The official land cover map for Santa Cruz was elaborated based on the 2018 product of the Climate Change Initiative (CCI) of the European Space Agency (ESA) [60]. This map was reclassified taking into account the portion of the area covered by each class; classes with a surface lower than 0.1% of the total study area were aggregated to the closest class. The resulting 10 classes are listed in Table 2.

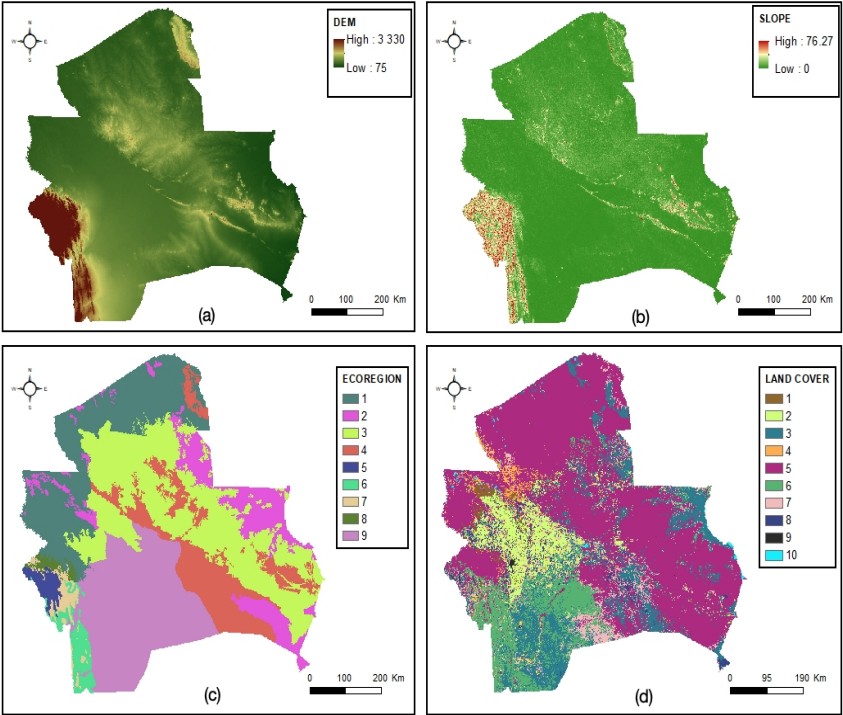

**Figure 6.** Predisposing factors: (**a**) DEM (Altitude, in meters), (**b**) Slope (in degrees), (**c**) Ecoregions, and (**d**) Land cover classes (see Table 2 for class descriptions).

**Table 2.** Ecological, land cover, and vegetation factors.

| Pred. Factors | #. Class | Area (ha) | Area (%) |
|---|---|---|---|
| Ecoregions | **1.** Southwestern Amazon forest | 7,095,085 | 19.42 |
| | **2.** Flooded Savanna | 3,855,600 | 10.55 |
| | **3.** Chiquitano Dry forest | 10,282,677 | 28.14 |
| | **4.** Cerrado | 4,768,190 | 13.05 |
| | **5.** Dry Inter-Andean forest | 504,133 | 1.38 |
| | **6.** Chaco Serrano | 934,197 | 2.56 |
| | **7.** Bolivian Tucuman forest | 535,884 | 1.47 |
| | **8.** Yungas | 261,457 | 0.72 |
| | **9.** Gran Chaco | 8,297,441 | 22.71 |
| Land cover | **1.** Cropland, rainfed | 499,512 | 1.37 |
| | **2.** Herbaceous cover | 2,476,316 | 6.77 |
| | **3.** Shrub or herbaceous cover, flooded, fresh/saline/brakish water | 5,517,848 | 15.09 |
| | **4.** Mosaic cropland (>50%) / Natural vegetation (tree, shrub, herbaceous cover) (<50%) | 919,420 | 2.51 |
| | **5.** Tree cover, broadleaved, evergreen, closed to open (>15%) | 19,873,239 | 54.33 |
| | **6.** Tree cover, broadleaved, deciduous, closed/open (15-40%) | 3,988,847 | 10.91 |
| | **7.** Mosaic natural vegetation (tree, shrub, herbaceous cover) / cropland (<50%) | 2,245,102 | 6.14 |
| | **8.** Grassland | 890,728 | 2.44 |
| | **9.** Bare areas/Urban areas/Sparse vegetation (>15%) | 46,897 | 0.13 |
| | **10.** Water bodies | 120,082 | 0.33 |

*3.3. Machine Learning Approach: Random Forest*

RF is an ensemble-learning algorithm based on decision trees [61]. As in general for ML-based approaches, RF is capable of learning from and making predictions on data by modeling the hidden relationships between a set of input and output variables. Inputs are the independent variables, or predisposing factors, while the outputs are the dependent variables, represented in the present case by the burned areas.

In more details, decision trees are supervised classifiers providing decisions at multiple levels and constituted by a root node and child nodes. At each node, decisions are performed based on training predictor values. The number of trees (*ntree*) to implement the model and the number of variables (*mtry*) randomly sampled as candidates at each split, are the only hyperparameters that need to be specified by the user. The algorithm generates *ntree* subsets of the training dataset, counting about two-thirds of the observations chosen by bootstrapping (i.e., random sampling with replacement). The remaining one-third of the observations (called "out-of-bag") are kept out and used to assess the prediction-error, allowing to optimize the values of the hyperparameters. For each subset, *mtry* variables are then randomly selected at each decision tree and, at each node, the best variable is assessed based on the minimization of the Gini impurity value. This last denotes the probability of classifying incorrectly an observation if it were classified randomly, according with the class distribution in the dataset. The prediction of new data is finally computed by taking, for classification problems, the maximum voting of all the trees. This value can be converted into a probabilistic output, by normalizing it over the number of iterations (i.e., *ntrees*). In this study, hyperparameters were set to 500 for *ntree* and 3 for *mtry*.

In addition, RF allows to assess the relative importance of each variable on the prediction. This is achieved by evaluating the Mean Decrease Accuracy, computed by estimating how much the tree nodes which use that variable reduce the mean square errors on the

out-of-bag. Moreover, the partial dependence plot gives a graphical depiction of the marginal effect of each variable on the class probability over different ranges of continuous or discrete values.

### 3.4. Model Validation

A well-established procedure to validate the outputs of a model in ML is to split the dataset into three-subsets, defined as training, validation, and testing. In more details, the training subset is needed to generate the model, which will be used to get predictions on new data. The ultimate purpose of the validation subset is the optimization of the hyperparameters of the model, performed by applying a trial and error process (i.e., by comparing the values predicted with the observed). Lastly, the testing subset provides an unbiased evaluation of the final model, allowing to assess its performance in making good predictions on new data, supposed to be drawn from the same distribution as the training data. Indeed, a good model has to give accurate predictions on previously unseen data and avoid under- and over-fitting.

Actually, the out-of-bag in RF can act for validation purposes. Nevertheless, when dealing with spatio-temporal environmental phenomena, the random selection of the data in the out-of-bag leads to a spatial issue of auto-correlation, meaning that most of the training and validation data will probably be close each other, holding similar characteristics. To overcome this, we selected the validation subset using a spatial k-fold cross-validation approach. This consists of splitting the original training dataset into k-folds, keeping out a fold at a time, training the model on the k-1 folds, and finally validating the model using the kept-out fold. The process is repeated k-times and the evaluation scores resulting from each folding is finally averaged. The use of k-folds assures a better spatial generalization of the final model. Here, we considered 5-folds, organized into spatial blocks of $100 \times 100$ km for the entire department of Santa Cruz, and of $50 \times 50$ km for the municipality of San Ignacio de Velasco. In the present study, the Area Under the Receiver Operating Characteristic (ROC) Curve (AUC) represents the evaluation score used as indicator of the goodness of the model in classifying areas more susceptible to burn. ROC is a graphical technique based on the plot of the percentage of correct classification (the true positives rate) against the false positives rate (occurring when an outcome is incorrectly predicted as belonging to the class "x" when it actually belongs to the class "y"). The AUC value lies between 0.5, denoting a bad classifier, and 1, denoting an excellent classifier.

To asses the predictive performance of the model, that is its ability to make a good prediction for future events, an independent testing subset was selected by splitting temporally the original dataset. The burned areas observed in the period of 2010–2016 were used in the training procedure, while the last three years of observations (2017–2019) were used as testing. To this end, we computed the percentage of the area with a probabilistic output value in a certain range, falling inside the burned areas in the testing dataset. Therefore, we can assume that the implemented model gives good prediction if it results in a higher percentage for high predicted values and a lower percentage for low predicted value.

### 4. Results and Discussions

#### 4.1. Wildfires Susceptibility Mapping in Santa Cruz

Outputs of RF are probabilistic values, expressing the probability of burning for each pixel under the assumption of a set of predisposing factors, namely altitude, slope, ecoregions, and land cover. These results provided the necessary information for the elaboration of a wildfire susceptibility map for the department of Santa Cruz (Figure 7). Probabilistic predicted values have been ranked into four classes, defined by equal intervals of 25% percentile ranges. It follows that the first quartile (i.e., the first interval) represents the 25% of the area with the lowest probability of burning, while the last quartile represents the 25% of the area with the highest probability of burning, and so on for the intermediate ranges. These range limits correspond different probabilistic output values (Prob. value), allowing a flexible interpretation of the results. It can be observed that the south-west

area results to be low susceptible to wildfires, as for a central band developing from north-west to south-east, which is lightly susceptible. On the other hand, very high classes of susceptibility, corresponding to a probabilistic predicted values higher that 0.7, are mostly located in the central and northern areas.

The main causes contributing to wildfire susceptibility in the department of Santa Cruz are the following:

- The high rates of deforestation that occur in the different municipalities. Four out of ten municipalities with the largest deforested areas in Bolivia until 2013 are located in the Obispo Santistevan province [62]. Similarly, according with more recent data [63], the municipality of San Ignacio de Velasco in the province of José Miguel de Velasco headed the list of the 25 Bolivian municipalities with the highest levels of deforestation between 2016 and 2018.
- The presence of large livestock properties known for burning large areas to enlarge pastures and agricultural lands. For instance, in the municipality of San Matías, located in the province Ángel Sandoval and corresponding to the second area most affected by the extreme fires of 2019, 75% of the burned areas can be imputed to the farming industry [50].
- High rates of wildfires initiated in neighboring countries, close to the border. For example, the Germán Busch province, located in the Bolivian Pantanal, has large areas with very high wildfire incidence due to the spreading of fires that start in Brazil. It was indeed verified that many of the fires that affected the National Park and the Integrated Management Area Otuquis originated in Brazil and then spread up to this area [64].
- Activities linked to the urban areas. For instance, in the Sara province, one of the main causes of wildfires is uncontrolled waste burning [62].

### 4.1.1. Model Validation and Performance Evaluation

The ROC curve was estimated on the validation dataset, selected based on the 5-fold spatial cross-validation procedure. This method is usually employed to determine the quality of model predictions. The accuracy index corresponds to the AUC, for which values close to 1 indicate that the resulting susceptibility map has an excellent accuracy. For the implemented model, it results in an AUC of 0.73, corresponding to a good performance.

Results of the predictive performance of the model are presented in Table 3. RF outputs were organized into four classes ranking, according to the quartile distribution (*Perc.*). In addition, the corresponding probabilistic output value is specified in the table (*Prob. value*). For each range, the area coincident with the testing polygons (*Testing BA*, corresponding to the years 2017, 2018, and 2019) is computed both as percentage and as absolute surface value (in hectares). Therefore, the field *Testing BA [%]* represents the percentages of the *Total area* included in the burned area detected in the testing dataset. For example, looking at the first row, the 25% of the *Total Area* with the lowest probability of burning (first quartile) corresponds to the output with a probabilistic value between 0 and 0.04 and it results that 6.77% of this areas is coincident with the burned area of the testing dataset. On the opposite extreme, 50.69% of the areas in the last quartile (>75%) (i.e., the 25% of the *Total Area* with the highest probability of burning included into a probabilistic values range between 0.7 and 1), is coincident with the burned area of the testing dataset (Figure 8). For each single testing period, values correspond to 63.54% (339,153 ha) in 2017, 56.16% (411,119 ha) in 2018, and 50.67% (1,687,505 ha) in 2019.

The last row informs us that for half of the *Total Area* with the highest probability of burning, 75.79% lies within the testing burned area (Figure 8). For each single testing period, this value corresponds to 84.36% (450,300 ha) in 2017, 81.12% (593,851 ha) in 2018, and 75.31% (2,508,189 ha) in 2019. It follows that the developed model displays the best predictive performance in 2017 and the worst in 2019.

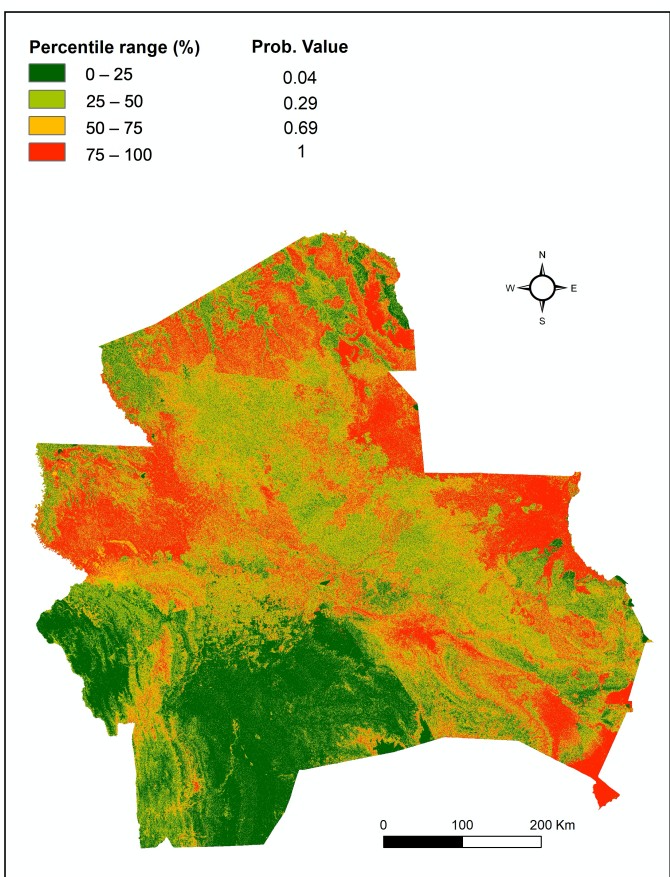

**Figure 7.** Wildfire susceptibility map of Santa Cruz.

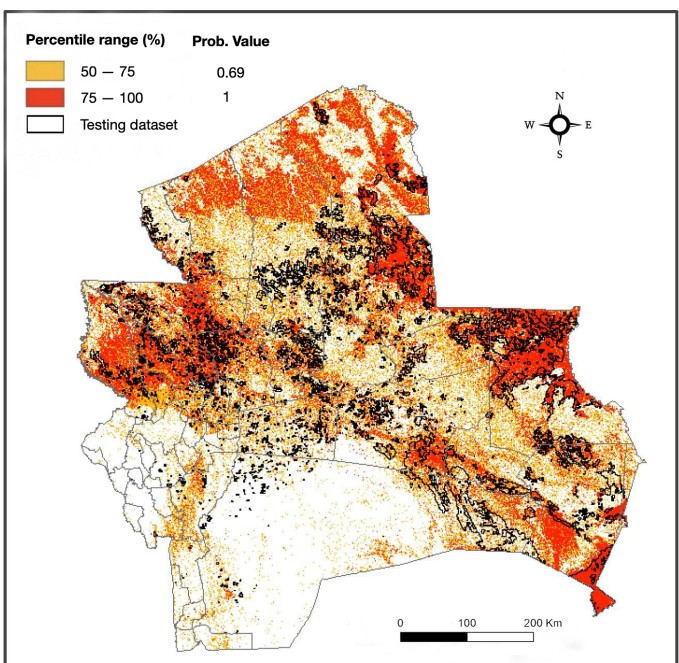

**Figure 8.** Performance evaluation of the RF outputs (Percentile range and corresponding Probability values) for the department of Santa Cruz, assessed by using the testing dataset (2017–2019 burned area).

**Table 3.** Model validation evaluated by computing the predicted burning area (BA) for different percentile classes (Perc.) falling within the testing burned areas, estimated globally (Testing BA), and for each single year (2017, 2018, 2019).

| Perc. | *p*-Value | Testing BA | | BA 2017 | | BA 2018 | | BA 2019 | |
|---|---|---|---|---|---|---|---|---|---|
| | | [%] | [ha] | [%] | [ha] | [%] | [ha] | [%] | [ha] |
| <25% | 0–0.04 | 6.77 | 281,460 | 4.7 | 25,142 | 5.7 | 41,546 | 6.7 | 223,272 |
| 25–50% | 0.04–0.3 | 17.4 | 725,102 | 10.9 | 58,315 | 13.2 | 96,672 | 18 | 598,737 |
| 50–75% | 0.3–0.7 | 25.1 | 1,043,996 | 20.8 | 111,147 | 25 | 182,732 | 24.6 | 820,684 |
| >75% | >0.7 | 50.7 | 2,108,121 | 63.5 | 339,153 | 56.2 | 411,119 | 50.7 | 1,687,505 |
| >50% | >0.3 | 75.8 | 3,152,117 | 84.4 | 450,300 | 81.1 | 593,851 | 75.3 | 2,508,189 |
| Total | | 100 | 4,158,679 | 100 | 533,757 | 100 | 732,089 | 100 | 3,330,198 |

### 4.1.2. Variable Importance Ranking

The variable importance ranking is shown in Figure 9. Based on the *Mean Decrease Accuracy*, the land cover results to be the most important variable, followed by ecoregions. This implies that the topology (i.e., altitude and slope) plays a secondary role in wildfires prediction in the area, compared with the physical coverage of the Earth's surface.

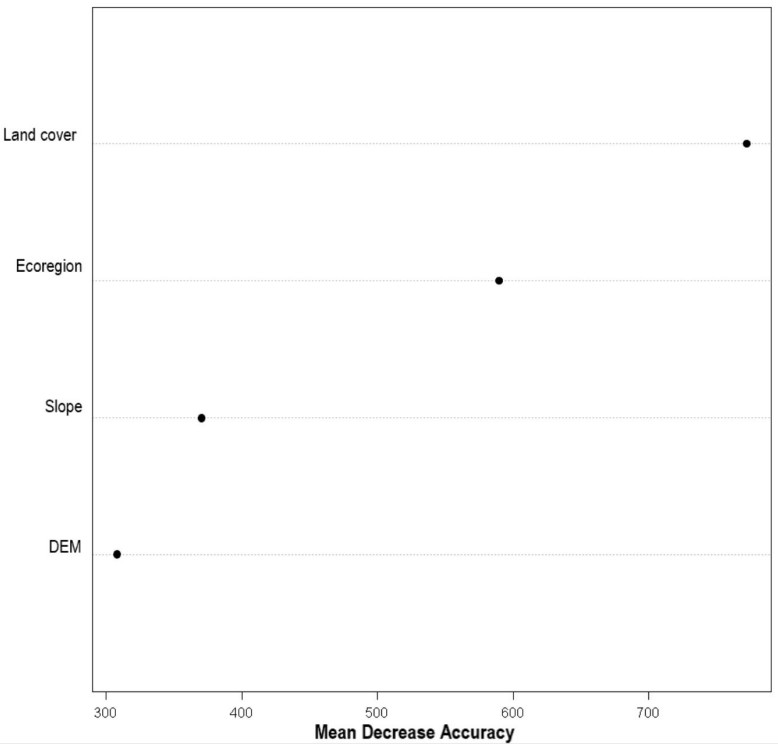

**Figure 9.** RF variable importance analysis for Santa Cruz based on the Mean Decrease Accuracy.

Three types of ecoregions are particularly prone to wildfires: Southwestern Amazon forest, Chiquitano dry forest, and flooded savanna. Indeed, they are associated to the three types of forests that proved to be the most affected by fires: Amazon forest, Chiquitano forest and Pantanal forest. For instance, previous studies demonstrated that, during the course of their evolution, the species of trees found in dry forests were exposed to fires of low to moderate severity [65,66]. Therefore, they have a limited capacity to recover naturally following the extensive high-severity fire of the last decades. On the other hand, the Cerrado ecoregion is structurally open, containing abundant fire-tolerant woody species [67]. Thus, for a good interpretation of the results performed by the present

analysis, it is important to take into account the different types of ecoregions and the land cover classes.

The partial dependence plot (pdp) allows to estimate, for each single factor, the relative probability of prediction success over different ranges of discrete values. This was evaluated for the two most important factors: land cover (Figure 10) and ecoregions (Figure 11).

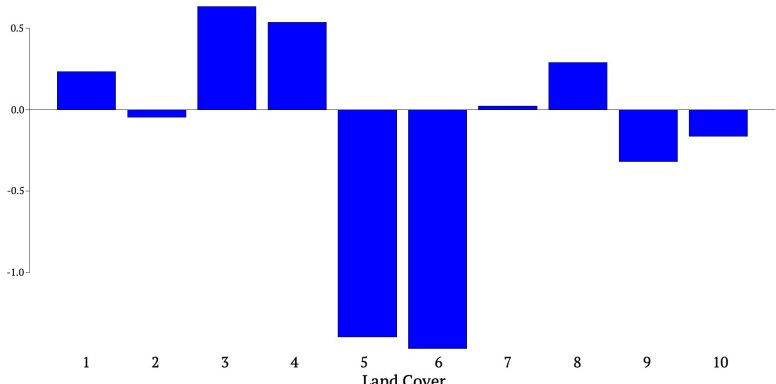

**Figure 10.** Partial dependence plot on the explanatory variable land cover. See Table for class descriptions.

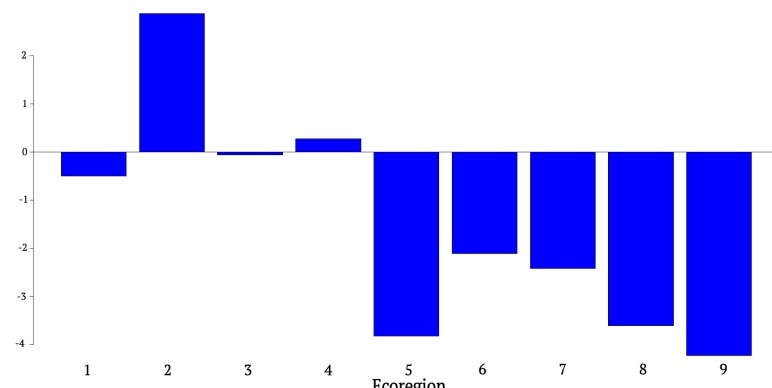

**Figure 11.** Partial dependence plot on the explanatory variable ecoregions. See Table 2 for class descriptions.

For land cover (Figure 10), the probability of predicting the observed burned areas is higher for the class shrub or herbaceous cover, flooded, fresh/saline/brakish water. Three kinds of forests belong in this category: Chiquitano forest, Pantanal forest, and Chaqueño forest. The former possessed a very high probability of burning as we could see in the susceptibility map. Both the Chiquitano and the Pantanal forests are under increasing agricultural pressure and intensifying cattle ranching activities, recurring to the heavy use of slash-and-burn practice which caused the invasion of fire-tolerant grass species [67]. According to the pdp, the less influential land cover type is tree cover, broadleaved, deciduous, closed/open (15–40%). This land cover type includes the following forests: Chaqueño, Yungas, and Tucumano-Boliviano forests.

The pdp for ecoregions (Figure 11) shows that the highest value is associated to flooded savanna. Indeed, this ecoregion is characterized by the presence of grassland and shrub or herbaceous cover, flooded, fresh/saline/brakish water. On the other hand, the less influential ecoregions are Gran Chaco and Dry-Inter Andean forest, which do not have a high incidence of wildfires. These results are also in accordance with the previous considerations on the relative importance of the various land cover types.

### 4.2. Results of the Model in San Ignacio De Velasco Municipality

The model developed for the entire department of Santa Cruz was then applied to the municipality of San Ignacio de Velasco, allowing a deeper analysis at the local scale, which implies a more homogeneous distribution of the predisposing environmental variables. Probabilistic outputs of the RF allowed to elaborate the wildfire susceptibility map (Figure 12). The 25% of the area with the highest probability of burning resulted to be more concentrated in the east and southeastern sectors. These areas are mainly characterized by the flooded savanna and Cerrado ecoregions, and by the land cover class shrub or herbaceous cover, flooded, fresh/saline/brakish water, which is very prone to wildfires. Indeed, slash-and-burn is a common practice in these areas, since they are the most exploited by humans for agriculture, livestock, and logging.

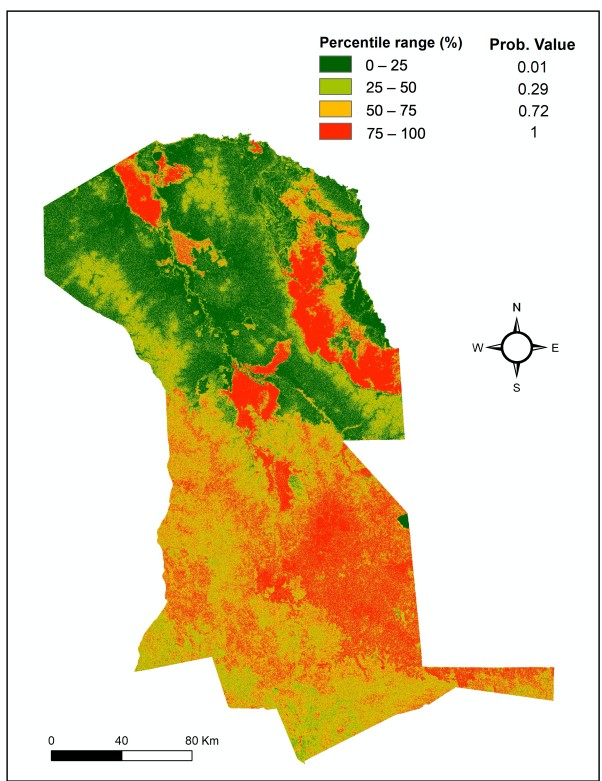

**Figure 12.** Wildfire susceptibility map of San Ignacio de Velasco municipality.

Predictions on the testing dataset are better in San Ignacio de Velascothe Figure 13 than in the entire department of Santa Cruz. Comparison between the predictive performance of RF in the two study areas is provided in Table 4. It results that, at municipality scale, 87.59% of the burned area within the testing dataset falls in the half of the area predicted with the highest probability of burning, compared to 75.79% at departmental scale. Looking at each single testing period, values are very high in 2017 (92.36%) and 2018 (91.15%), while in 2019 the performance of the model is less good (86.29%). Taking in consideration the last quartile (>75%), these values dropped of about 30%, with an average value of 56.51% considering the entire testing dataset, compared to 50.69% at the departmental scale.

**Table 4.** Comparison between the predictive performance of the model by quantile ranking range classification for San Ignacio de Velasco and Santa Cruz.

| Study Area | Perc. | Testing Dataset | | Year 2017 | | Year 2018 | | Year 2019 | |
|---|---|---|---|---|---|---|---|---|---|
| | | [%] | [ha] | [%] | [ha] | [%] | [ha] | [%] | [ha] |
| **S. Ignatio de Velasco** | >50% | 87.59 | 667,850 | 92.36 | 134,816 | 92.15 | 207,346 | 86.29 | 448,879 |
| | >75% | 56.51 | 430,868 | 66.52 | 97,095 | 69.14 | 155,577 | 52.48 | 272,994 |
| **Santa Cruz** | >50% | 75.79 | 3,152,117 | 84.36 | 450,300 | 81.12 | 593,851 | 75.31 | 2,508,189 |
| | >75% | 50.69 | 2,108,121 | 63.54 | 339,153 | 56.16 | 411,119 | 50.67 | 1,687,505 |

The best performance of the model in the municipality of San Ignacio de Velasco, compared with the entire department of Santa Cruz, is confirmed by the results of the AUC estimated from the ROC curve (Figure 14). These indicate an accuracy of 0.8 at the municipality scale, which is a higher value than the one obtained at departmental scale (0.73).

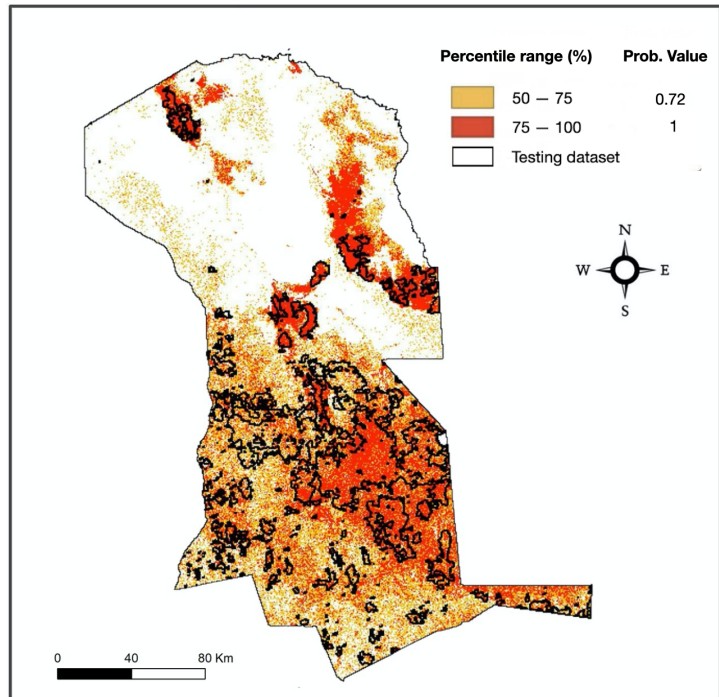

**Figure 13.** Performance evaluation of the RF outputs (Percentile range and corresponding Probability values) for the municipality of San Ignacio de Velasco, assessed by using the testing dataset (2017-2019 burned area).

Results of the analysis of the variable importance ranking are presented in Figure 15. As for the entire department of Santa Cruz, ecoregions and land cover are, respectively, the first and the second most important variable, while slope and DEM are not very reliable factors. The dominant ecoregions in the area were also the most affected by wildfires: southwestern Amazon forest, Chiquitano Dry forest, and flooded savanna. Furthermore, the most diffuse types of forest are the Amazon forest and the Chiquitano forest, for which a natural recovery after an extensive and severe fire is very difficult.

The pdp of land cover (Figure 16) reveals that the most important class for the prediction of wildfires is shrub or herbaceous cover, flooded, fresh/saline/brakish water. Once again, this is in accordance with the results obtained for Santa Cruz. Indeed this type of land use is employed for agriculture, livestock, and logging, which are related to high rates of deforestation within the municipality. The less influencing land cover type according to the pdp is tree cover, broadleaved, evergreen, closed/open (>15%), even if this prevails in the municipality area. The reason behind can be that, since this cover is widely present, the

model is not able to distinguish between the presence and absence of wildfires here. As regards the pdp for ecoregions (Figure 17), the graphic shows that flooded savanna is the most influential type for the predictive success of the model. The less important one is, in this case, the southwestern Amazon forest, probably because of the high wetness index characteristic of the original vegetation [68].

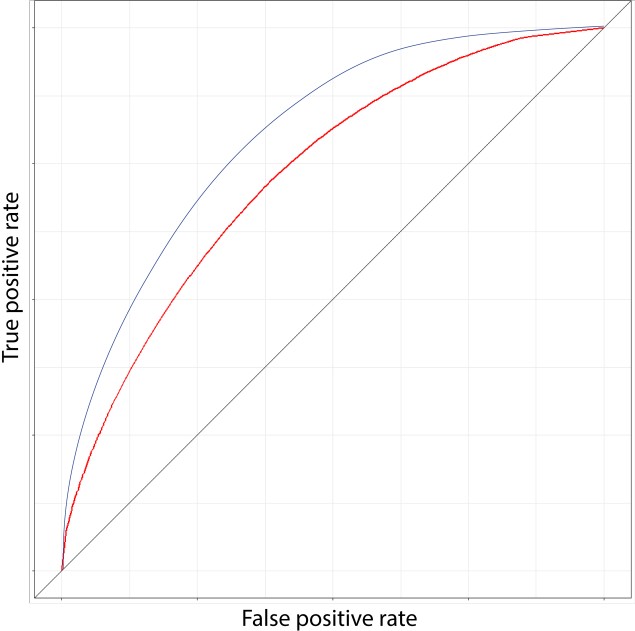

**Figure 14.** ROC curves for Santa Cruz (in red, AUC = 0.73) and for San Ignacio de Velasco (in blue, AUC = 0.8).

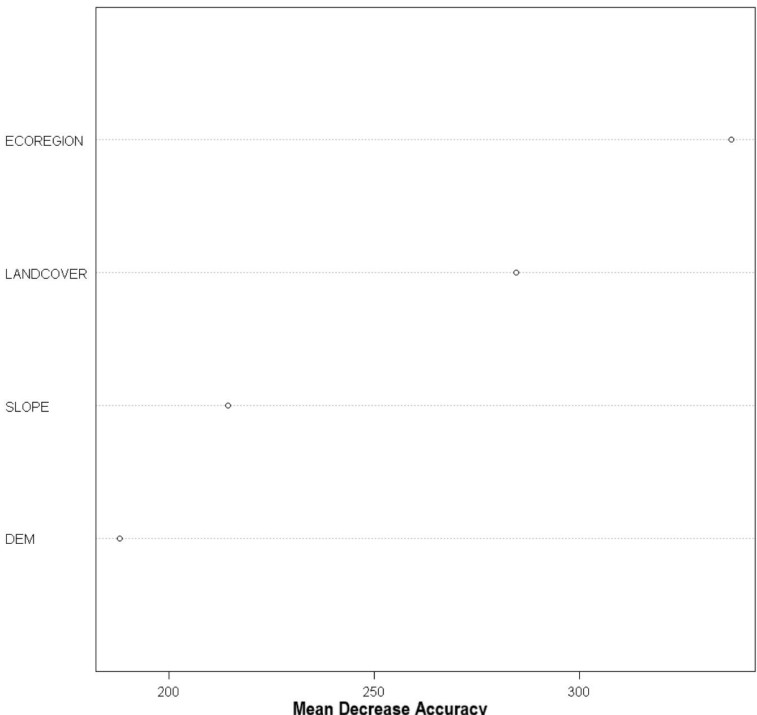

**Figure 15.** RF variable importance analysis for San Ignacio de Velasco based on the Mean Decrease Accuracy.

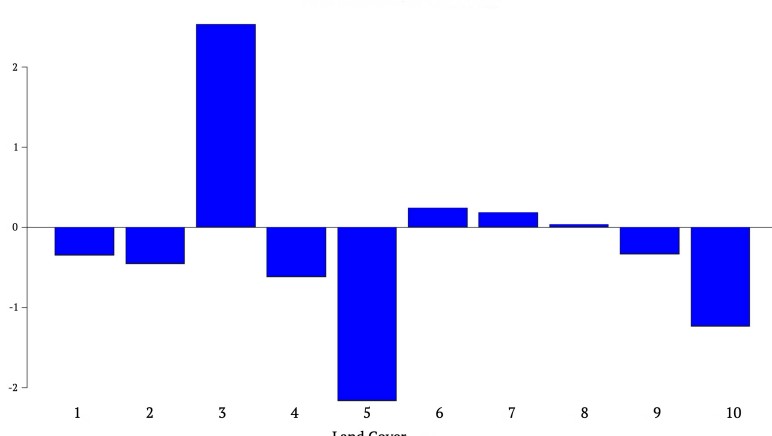

**Figure 16.** Partial dependence plots on the explanatory variable land cover. See Table 2 for class descriptions.

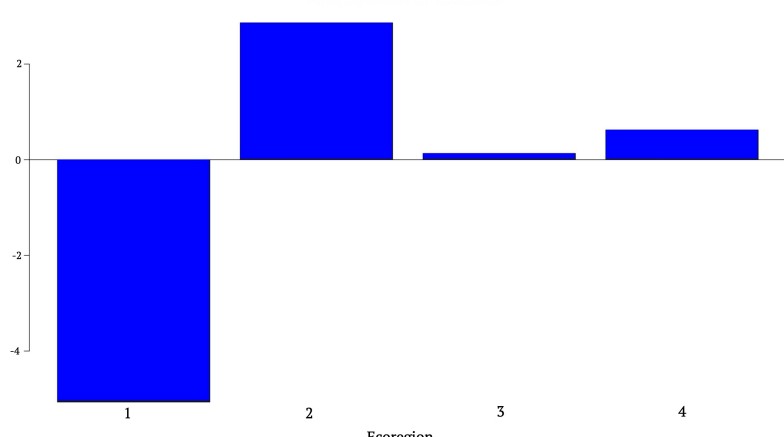

**Figure 17.** Partial dependence plots on the explanatory variable ecoregions. See Table 2 for class descriptions.

## 5. Conclusions

The central-South American forest is one of the major fire-prone areas worldwide [42,43]. Land use management and climate changes have caused wildfires to become more frequent and extended in the recent years [69,70]. In Bolivia, the department of Santa Cruz was hit by extreme fires in 2019 and globally this area accounts for more than two-thirds of the total wildfires in the country [71,72]. Despite Bolivia being in the list of the top countries for the expected annual burned forest area, the literature on wildfires is quite limited, also because of the scarcity of available data, resources, and informatization. The present study contributes to fill this gap, thanks to the implementation of an accurate dataset of burned area that occurred in the entire department of Santa Cruz in the period of 2010–2019. In addition, environmental variables that can favor the wildfire ignition and spread (related to topography, land cover, and ecoregions) have been collected and processed as digital spatial information. This accurate dataset allowed implementing a machine learning-based model, using the Random Forest algorithm, providing as output the probability of burning for each single unit-area. The model was applied both at the departmental and at the municipality scale, considering the smaller and more homogeneous area of San Ignacio de Velasco.

The area under the ROC curve (AUC) was computed for validation purposes, based on a 5-fold cross-validation procedure. In addition, an independent testing subset was selected by splitting temporally the original dataset, allowing to assess if the model makes good prediction on future events. As a result, we obtained a probabilistic output which

allowed elaborating the susceptibility maps for the two study areas. The model validation gave good results, that improved at the municipality scale (AUC = 0.8) compared to the departmental scale (AUC = 0.73). Likewise for the prediction on the testing dataset (2017–2019), for which it results that more than 50% of the burned area coincides with the last quartile on the susceptibility map for both the study areas. It is worth noting that in the first two years (2017 and 2018) about 60% of the burned area was correctly predicted for the department of Sanza Cruz, and that this value increased from 66.52% to 69.14% for San Ignacio de Velasco. Indeed, the extreme wildfire events of 2019 made the prediction of the model less accurate. In this regard, it must be stressed that the testing subset is supposed to be drawn from the same distribution as the training data. It follows that, if its distribution differs from the observations used to train the model (e.g., extreme events or outside the norm), the model will results in weaker predictions. We must mention here that, among the two million wildfires registered worldwide every year in the recent period, only a few become extreme events [73,74]. Extreme wildfires cause substantial damages and often result in civilian and firefighter fatalities since they tend to overwhelm suppression capabilities. Modeling and predicting these extraordinary events is very challenging since their extreme behavior and impact result from the complex interaction with atmospheric processes and climatic conditions (such as heat waves, long dry periods, very strong and variable winds) and local conditions (e.g., low fuel moisture content, landscape connectivity, inadequate initial attack, poor preparedness, and vulnerable communities). Thus, even if advanced wildfire susceptibility models are promising tools for wildfire assessment and prediction, future work should address the development of prediction mapping for extreme wildfire events.

The detailed investigation of the relative importance of each categorical class belonging to the variables ecoregions and land cover reveals that "flooded savanna" and "shrub or herbaceous cover, flooded, fresh/saline/brakish water" are the two classes most related with wildfires. This important outcome confirms recent findings, that seasonally wet and dry climate, coupled with hydrologic controls on the vegetation, create in this ecoregion conditions favorable to the ignition and spreading of large wildfires during the driest period, when the biomass is abundant [75,76]. The occurrence of large fires, initiated by slash-and-burn practice getting out of control, is predicted to increase in the near future and the development of new tools for fire risk assessment and reduction is thus needed.

To conclude, it is worth to underline that wildfires are likely to become more dominant in Santa Cruz because of the vicious cycle linking the current slash-and-burn deforestation practices, the rapid frontier expansion, and longer drought periods. In a broader sense, these harmful conditions could be further exacerbated by the increased moisture stress acting in different forests of great global relevance such as the Amazon, Chiquitano, and Pantanal coupled with the spreading use of fire in these areas. Therefore, it is necessary to dispose of key tools for wildfire prevention-planning programs aiming to reduce human, ecological, and material losses. In addition, to take full advantage of machine learning for the mapping and assessment of natural disaster, it is of paramount importance to develop an accurate and updated digital geospatial dataset and more effort has to be done in this regard.

This study proved that: (i) Random Forest can successfully be employed for prediction mapping and to asses wildfire predisposing factors; (ii) it is possible to implement a simple but powerful model even for a country, such as Bolivia, with poor resources in terms of data-availability and informatization.

**Author Contributions:** Conceptualization: M.T. and P.F.; data curation, M.B.S.; methodology, M.T.; formal analysis, M.B.S.; investigation, M.B.S. and M.T.; project administration, M.T. and P.F.; supervision, M.T. and A.M.; validation, M.T. and A.M.; visualization, M.B.S.; writing—original draft, M.B.S.; writing—review and editing, M.T. All authors have read and agreed to the published version of the manuscript.

**Funding:** This research received no external funding.

**Institutional Review Board Statement:** Not applicable.

**Informed Consent Statement:** Not applicable.

**Data Availability Statement:** The MODIS burned area product is available for download in either HDF, GeoTIFF, or Shapefile format from the University of Maryland fuoco SFTP (formerly FTP) server. https://modis-fire.umd.edu/guides.html accessed on 18 May 2021.

**Acknowledgments:** This study is part of the Master's thesis of Marcela Bustillo Sánchez. Authors acknowledge the "Simon I. Patiño Foundation" (https://www.fondationpatino.org/en/, accessed on 18 May 2021) for founding her scholarship, aimed at supporting her Master's degree at the University of Lausanne (Switzerland).

**Conflicts of Interest:** The authors declare no conflict of interest. The founders had no role in the design of the study; in the collection, analyses or interpretation of data; in the writing of the manuscript, or in the decision to publish the results.

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
