# Peer review of "Spatial Assessment of Wildfires Susceptibility in Santa Cruz (Bolivia) Using Random Forest"

_geosciences, doi:10.3390/geosciences11050224_

Round 1
Reviewer 1 Report
Dear authors. Thank you for your interesting and important article, which I have read and reviewed with great pleasure.
However, I have found several aspects that need to be significantly improved in order for your article to be considered for publication; the main aspects concern: (i) English language, (ii) methodological aspects not fully described, (iii) poor image quality.
Detailed reviews and comments can be found in the attached pdf.
Best regards

Author Response
We thanks the Reviewer for the carefully revision and comments.
All the minor changes have been implemented directly on the uploader new version. These includes:
- We used comma (,) to separate thousands and dot (.) to separate decimal. We changed the separators everywhere in the manuscript: in the text, figures and tables.
- “When entering an acronym for the first time, we used capital letters in the extended entry”: we checked all the items in the text and we did it.
- We revised all the bibliographic references in the manuscript, especially the one highlighted by the reviewer in the Introduction. All the citation referred to Machine Learning application to wildfires in the Introduction (Line 36-60) have been improved and presented in a more consistent way (see References from [25 – 40])
- Line 81:“The main cause of fire ignition in the area is human-made, principally due to the commune practice of slash-and-burn”: we added two references to support this statement, as suggested by the Reviewer.
- We added a Figure with the two ROC curves (Figure 14 ) as suggested by the Reviewer
- In “Conclusions” (Line 462-466) we added two references for each one of the three sentence, as proposed by the Reviewer.
In addition, we improved different aspect as mentioned by Reviewer #1:
- We carefully revised the English language;
- Methodological aspects: Figure 5 (“Workflow of the methodology employed in this study”) is now correctly displayed and this help to better understand and follow each methodological step implemented in the present study; moreover, we added more details on MODIS burned area dataset ((Line 162-166, “Data”);
- We improved the quality of all the Figures.
Reviewer 2 Report
The study presented in this paper is interesting and very useful for the management of wildfire risk.
The abstract should have three noticeable parts: theory, methodology and results. And that is not the case. I advise that the abstract be improved in order to give more complete information to the reader.
The introduction provide sufficient background and include relevant references. However, more recent references could be added.
A good characterization of the study area is made. In some cases, you could refer to the most recent bibliography and present data from 2020. The same comments for chapter 3.
Chapter 4 presents the results and discusses them. The results are interesting and are presented in a practical and easy to interpret way. Figures 10 and could be improved in order to allow a better reading of the bars caption.
The conclusions presented show the importance of the work done. You could present future work, which is always recommended and appreciated.
The references presented are recent and solid. Some additional recent reference would be helpful.
Some additional notes:
- The quality of the images is not very good. This can and should be improved. The image quality may have been reduced in the pdf recording. Confirm that, please.
- The data presented in figures 2, 3 and 4 do not include the year 2020. Why?
- Something is wrong in figure 5.
- Why do you use the comma as a decimal separator?
Author Response
We thanks the Reviewer for the carefully revision and comments.
Reviewer: The abstract should have three noticeable parts: theory, methodology and results. And that is not the case. I advise that the abstract be improved in order to give more complete information to the reader.
Authors: we completely change the abstract as follows: “Wildfires are expected to increase in the near future, mainly because of climate changes and land use management. One of the most vulnerable are in the world is the forest in central- South America, included Bolivia. Despite this country is highly prone to wildfires, literature is rather limited here. To fill this gap, we implemented a dataset including the burned area occurred4in the region of Santa Cruz in the period 2010-2019, and the digital spatial data describing the predisposing factors (i.e. topography, land cover, ecoregions). The main goal was to develop a model, based on Random Forest, which probabilistic outputs allowed to elaborate wildfires susceptibility maps. The overall accuracy was finally estimated using 5-folds cross validation. In addition, the last three years of observations were used as testing dataset to evaluate the predictive performance of the model. The quantitative assessment of the variables revealed that “Flooded savanna” and “Shrub or herbaceous cover, flooded, fresh/saline/brakish water” are respectively the ecoregions and land cover classes with the highest probability of predicting wildfires. This study contributes to the development and validation of an innovative mapping tool for fire risk assessment, implementable at regional scale in different areas of the globe.”
Reviewer: The introduction provide sufficient background and include relevant references. However, more recent references could be added.
Authors: We included more recent references not only in the Introduction but also in Chapter 2 and 3, as suggested by the Reviewer. We added a total 20 more references, all of them published after 2017.
Reviewer: A good characterization of the study area is made. In some cases, you could refer to the most recent bibliography and present data from 2020. The same comments for chapter 3.
Authors: See above for references. As concern data from 2020, MODIS product was available only up to 2019 at the time the analyses has been carried out.
Reviewer: Chapter 4 presents the results and discusses them. The results are interesting and are presented in a practical and easy to interpret way. Figures 10 and could be improved in order to allow a better reading of the bars caption.
Authors: image quality has been improved
Reviewer : The conclusions presented show the importance of the work done. You could present future work, which is always recommended and appreciated.
We thank the Reviewer for this suggestion which gave us the possibility of introducing the problem of modeling and predicting extreme wildfire events. To this end, we added the following paragraph (Lines 495-505) : “We must mention here that, among the two million wildfire events registered in the recent period worldwide every year, only few become extreme events4 [73,74]. Extreme wildfires cause substantial damages and often result in civilian and firefighter fatalities since they tend to overwhelm suppression capabilities. Modeling and predicting these extraordinary events is very challenging since their extreme behavior and impact result from the complex interaction with atmospheric processes and climatic conditions (such as heat waves, long dry periods, very strong and variable winds) and local conditions (e.g. low fuel moisture content, landscape connectivity, inadequate initial attack, poor preparedness, and vulnerable communities). Thus, even if advanced wildfire susceptibility models are promising tools for wildfire assessment and prediction, future work should address to the development of prediction mapping for extreme wildfire events.”
Reviewer : The references presented are recent and solid. Some additional recent reference would be helpful.
Authors: See above for added references
To answer to the additional notes, we improved the quality of all the images and we changed the separators everywhere in the manuscript (in the text, figures and tables). As mentioned above, MODIS product was available only up to 2019 at the time the analyses has been carried out, which was part of the master thesis of Marcela Bustillo.
Reviewer 3 Report
The manuscript submitted by Bustillo Sancez et al. “Spatial assessment of wildfires susceptibility in Santa Cruz (Bolivia) using Random Forest” presents an application of a machine learning approach (Random Forest) for deriving a wildfire susceptibility map in the department of Santa Cruz, Bolivia. Study area is important occupying more than one third of the entire national territory; the region is covered mainly by a mosaic of wet / dry tropical forests and herbaceous cover of savannas.
The approach, developed by some co-authors for Liguria region (Italy), is here applied in a different ecosystem where the information on the importance/role of wildfires is less recognized and producing a susceptibility map can contribute to increase awareness on wildfires risk and suppression.
The paper is interesting and well presented.
Comments
Seasonality of wildfires has been suggested as an important factor for savanna type ecosystems (Govender et al., J. Applied Ecology, 43,4, 2006; N’Dri et al., Fire Ecol. 14, 5, 2018) and one of main wildfires feature impacting the environment (emissions into the atmosphere, equilibrium in vegetation cover, etc.). Although susceptibility map is static, this further aspect of fire seasonality should be mentioned in the Introduction.
Conclusions (Lines 486-490) The weaker predictions of the model for the year 2019 are justified “(e.g. extreme events or outside the norm)“. However from Fig. 2 and Fig. 3 also the year 2010, part of the training, presents an exceptional/extreme value of burned area. This concept on model limitations requires some more comments since the Introduction and when presenting data of Burned areas.
Burned areas are derived from MODIS wildfire product (MCD64): in addition to the reference (44) you should include the web site from where you downloaded the data.
MODIS Burned areas with a spatial resolution of 500-m were finally rescaled at 100 meters. Please specify the technique used for rescaling.
Line 254-256 “ … the training dataset, counting about one-third ….The remaining one-third “ It is not clear what is the second “one-third”
Table 2: it would be interesting to add a column with percentage of area; this information can give immediate impression of importance of a specific ecoregion or land cover class
Figure 5 illustrating the methodological workflow is not visible (it appears only text as “workflow.png workflow.pdf workflow.jpg …”
Figure 9. and Figure 14. Can be easily reduced in size without losing clarity
Minor remarks
Line 24 to limi the disastrous effects thayà to limit the disastrous effects they
Line 157 Burned area were à Burned areas were
Table 3. Model validation evaluated by computing the the predicted
> 25% à < 25%
Line 411 have a high incidence of wildres à wildfires
- Pacheco, P.; Mertens, B. ….. Bois et ForÃat des Tropiques à Bois et Forêts
Author Response
Reviewer: Seasonality of wildfires has been suggested as an important factor for savanna type ecosystems (Govender et al., J. Applied Ecology, 43,4, 2006; N’Dri et al., Fire Ecol. 14, 5, 2018) and one of main wildfires feature impacting the environment (emissions into the atmosphere, equilibrium in vegetation cover, etc.). Although susceptibility map is static, this futureer aspect of fire seasonality should be mentioned in the Introduction.
Authors: We agree with the Reviewer and indeed seasonality is a factor that need to be taken into consideration especially in area where more than one fire season (normally two: spring-summer and autumn-winter) are present, as the case of the previous study carried out by the Authors for Liguria region in Italy. Nevertheless, in the case of Bolivia typically there is only one fire season which support our choice. To better explain this concept we added the following paragraph (Lines 212-125): “Although forest in this region is exposed to a marked seasonality, and hence it is susceptible to changes in fire regimes, we did not considered this variability in the present study. Indeed, since typically there is only one fire season that hits during the frame-period coinciding with the driest months of the year, monthly burned area have been aggregated on yearly basis.”
Reviewer: Conclusions (Lines 486-490) The weaker predictions of the model for the year 2019 are justified “(e.g. extreme events or outside the norm)“. However from Fig. 2 and Fig. 3 also the year 2010, part of the training, presents an exceptional/extreme value of burned area. This concept on model limitations requires some more comments since the Introduction and when presenting data of Burned areas.
Authors: We thank the Reviewer for this suggestion which give us the possibility of introduce the problem of modeling and predicting extreme wildfire events. To this end, we added the following paragraph (Lines 495-505): “We must mention here that, among the two million wildfire events registered in the recent period worldwide every year, only few become extreme events4 [73,74]. Extreme wildfires cause substantial damages and often result in civilian and firefighter fatalities since they tend to overwhelm suppression capabilities. Modeling and predicting these extraordinary events is very challenging since their extreme behavior and impact result from the complex interaction with atmospheric processes and climatic conditions (such as heat waves, long dry periods, very strong and variable winds) and local conditions (e.g. low fuel moisture content, landscape connectivity, inadequate initial attack, poor preparedness, and vulnerable communities). Thus, even if advanced wildfire susceptibility models are promising tools for wildfire assessment and prediction, future work should address to the development of prediction mapping for extreme wildfire events.”
Reviewer: Burned areas are derived from MODIS wildfire product (MCD64): in addition to the reference (44) you should include the web site from where you downloaded the data.
We did it
Reviewer: MODIS Burned areas with a spatial resolution of 500-m were finally rescaled at 100 meters. Please specify the technique used for rescaling.
Authors: Indeed we did not rescaled this raster, but we worked at a spatial resolution on 100m that is the same of the DEM. We corrected in the manuscript.
Reviewer: Line 254-256 “ … the training dataset, counting about one-third (it’s two-third) !!!….The remaining one-third “ It is not clear what is the second “one-third”
Authors: Thanks for notice this error: indeed the algorithm generates ntree subsets of the training dataset, counting about two-third of the observations chosen by bootstrapping, and not one-third. We corrected in the manuscript.
Reviewer: Table 2: it would be interesting to add a column with percentage of area; this information can give immediate impression of importance of a specific ecoregion or land cover class
Authors: We thanks the Reviewer for this suggestion and we added a column with the percentage Area on Table 2
Reviewer: Figure 5 illustrating the methodological workflow is not visible (it appears only text as “workflow.png workflow.pdf workflow.jpg …”
Authors: Figure 5 (“Workflow of the methodology employed in this study”) is now correctly displayed.
Minor remarks
Figure 9. and Figure 14. Can be easily reduced in size without losing clarity.
Line 24 to limi the disastrous effects thayà to limit the disastrous effects they.
Done
Line 157 Burned area were à Burned areas were
Done
Table 3. Model validation evaluated by computing the the predicted
> 25% à < 25%
Done (thanks!)
Line 411 have a high incidence of wildres à wildfires
Done
- Pacheco, P.; Mertens, B. ….. Bois et ForÃat des Tropiques à Bois et Forêts
Round 2
Reviewer 1 Report
Dear authors,
Thank you for considering and responding to all my revisions. In my opinion, the article is now ready for publication.
Best regards